# Evaluation of the Implementation Effect of the Ecological Compensation Policy in the Poyang Lake River Basin Based on Difference-in-Difference Method

**Yu Lu [1,2], Fanbin Kong [1,2], Luchen Huang [3], Kai Xiong [4,*], Caiyao Xu [1,2,*] and Ben Wang [4]**

1   College of Economics and Management, Zhejiang Agriculture & Forest University, Hangzhou 311300, China; luyu_0802@163.com (Y.L.); kongfanbin@aliyun.com (F.K.)
2   Research Academy for Rural Revitalization of Zhejiang Province, Zhejiang A&F University, Hangzhou 311300, China
3   School of Economics, Jiangxi University of Finance and Economics, Nanchang 330013, China; 2202021580@stu.jxufe.edu.cn
4   Nanchang Institute of Technology, Nanchang 330099, China; 2015984555@nit.edu.cn
*   Correspondence: xiongkaing@hotmail.com (K.X.); xucaiyao@zafu.edu.cn (C.X.); Tel.: +86-150-7099-8907 (K.X.)

**Abstract:** Watershed environments play an important supporting role in sustainable high-quality economic development in China, but they have been deteriorating. In order to solve environmental problems in the Poyang Lake River Basin brought about by economic development, the Jiangxi Provincial Government promulgated relevant river basin protection policies in 2015. However, after several years of this policy, the specific effects of its implementation are a matter of general concern to the government and academic circles. After years of policy implementation, the implementation effect of the watershed ecological compensation policy needs to be evaluated. Based on 4248 observations from the Jiangxi and Hunan Provinces, we adopt the difference-in-difference method to analyze the impact of the ecological compensation policy on the Poyang Lake River Basin. The empirical results show that the ecological compensation policy has a significant effect on water-quality improvement. Water quality in the upstream area is better than that in the downstream area; areas with small administrative areas have a smaller population, which in turn leads to better water quality in the river basin; and the higher the per capita GDP, the worse the water quality. Our results highlight the need for the following policy improvements: ecological priority, customizing measures to local conditions, tracing the main body, and strengthening supervision.

**Keywords:** difference-in-difference method; Poyang Lake River Basin; ecological compensation; effect of policy implementation





## 1. Introduction

With more than 40 years of reform and opening the economy, China has achieved remarkable economic growth [1]. In 2019, China's GDP reached US $12242.776 billion with 6.1% annual growth rate, ranking second internationally with the United States being first [2]. However, due to the rapid economic development, environmental problems have increased [3], with quality deterioration of the ecological environment being the most prominent. In order to effectively alleviate and curb further quality deterioration of the ecological environment, General Secretary Xi Jinping clearly stated in November 2012 during the eighteenth National Congress of the Communist Party of China that we should vigorously promote the construction of an ecological civilization [4], strive to build a beautiful China, and realize a sustainable development of the Chinese nation [5]. In September 2013, General Secretary Xi, while in Nazarbayev University, Kazakhstan, stated "beautiful scenery, golden hill and silver mountain", highlighting the importance of green and sustainable development [6].

Since then, China's Central Committee, led by General Secretary Xi Jinping, has made the protection of the ecological environment a priority to ensure the sustainable development of China's economy and society [7]. Located in the Jiangxi Province, the Poyang Lake River Basin is the largest freshwater lake basin in China and has made important contributions not only to China's but also the world's economic development [8]. However, in recent years, there have been some ecological and environmental problems, such as a decline in water quality and a sharp decrease in watershed area [9]. The watershed ecological compensation mechanism is a comprehensive economic policy that promotes the internalization of external costs of water-environment protection and water pollution, improves the inter-regional collaboration on water-environment protection, and ensures coordinated upstream and downstream governance within the basin [10]. Therefore, in order to effectively protect the ecological environment in the Poyang Lake River Basin, in November 2014, the National Development and Reform Commission, the Ministry of Finance, the Ministry of Land and Resources, the Ministry of Water Resources [11], the Ministry of Agriculture and the State Forestry Administration officially approved the *implementation plan for the construction of an ecological civilization pilot zone in Jiangxi Province* (hereinafter referred to as the "plan"). In November 2015, the Jiangxi Provincial People's government issued and distributed the *ecological compensation measures for river basins in Jiangxi Province* (for Trial Implementation) in order to implement the contents of the plan and build the ecological compensation mechanism in the Poyang Lake River Basin [12]. The promulgation of the policy that aims to improve the water resources in the Poyang Lake River Basin is of great historical significance, and means that China has taken the lead in carrying out the pilot project of river-basin ecological compensation policy in Jiangxi Province [13]. However, it has been several years since the promulgation of the policy, and there are several issues to be addressed: What is the implementation effect of the ecological compensation policy? Are there differences among different regions in the implementation effect of the ecological compensation policy? Existing research cannot address the above questions. Based on the above, this study uses the water-quality monitoring data from the Poyang Lake River Basin in Jiangxi Province and the Dongting Lake River Basin [14] in Hunan Province from 2013 to 2018 to analyze the implementation effect of the ecological compensation policy.

## 2. Literature Review

Improving the quality of the ecological environment in river basins has always been an important issue for academia and governments around the world [15]. Among the various environmental protection measures, the construction of a watershed ecological compensation mechanism is considered an important measure towards improving basin environments [16]. The payment for cross-boundary ecosystem services is mainly based on the compensation between upstream and downstream or between regions in a market-oriented way [17]. As an example, research conducted jointly by Russia and Lithuania concerning Kulun Lagoon found that the improvement of the water environment depends on the support of high-level government officials and the dedicated cooperation between departments [18]. An effective way to reduce pollution is to introduce water-quality control policies by investigating the cost of transboundary pollution control and water quality [19]. In some Asian countries represented by China, the ecological compensation mechanism is mainly constructed by administrative means [20]. Since 2015, the Fujian and Jiangxi Provinces have taken the lead in introducing river-basin ecological compensation policies in order to significantly improve the ecological environment in the river basins [21]. Some scholars believe that the result of implementing the ecological compensation policy by administrative order is not ideal [22], mainly because the government's implementation is relatively inefficient and their compensation funds limited [23]; hence, the compensation effect cannot have great impact. However, some other scholars believe that the implementation of the ecological compensation policy led by the government can lead to significant improvement of the ecological environment in the basin [24]. Research on

ecological compensation mainly focuses on the construction of a theoretical system, the exploration of the mechanism, the compensation object, the compensation method [25], and the determination of compensation payment [26]. Few studies involve also post-compensation stage research, such as research on the performance evaluation of existing compensation projects, and especially research on the lack of performance evaluation of standard economic paradigms.

Several studies on the evaluation of the effect of watershed ecological compensation policies were initiated in recent years. Xu and Li conducted an empirical study on ecological compensation financial projects in Liaodong mountain areas and found that some counties, after the implementation of such policy, significantly impacted ecological performance [27]; however, some other counties did not. Qu et al., based on an empirical study on the implementation effect of ecological compensation policy in the Chishui River Basin of Guizhou Province, found that the implementation of an ecological compensation policy can effectively promote the improvement of the ecological environment of the local basin [28]. Wang et al. conducted an empirical analysis of the implementation effect of ecological compensation policy in Xin'an River Basin and found that the implementation of the policy was generally good, but the promotion effect on regional economic and social development was not significant [29]. Therefore, it is necessary to design and implement the policy while accounting for improvements of the policy objectives and system. At the same time, there are five common methods to evaluate the policy effect of the watershed ecological compensation policy. The first is the *instrumental variable method* (IV). This method deals with endogenous problems caused by standard econometrics [30]; however, this method has two main disadvantages: one is the choice of instrumental variables and the other is that it may not account for the heterogeneity of research objects. The second is the *breakpoint regression* (RD). Breakpoint regression is a quasi-experimental method similar to randomized controlled trials [31]. The main idea is that when the value of the sample individual's key variable exceeds the critical value, the individual will accept policy intervention; until then, the individual will not accept policy intervention [32]. The third is the *propensity score matching method* (PSM). This is a non-experimental method, which is an approximate experimental method with data from experimental and control groups that are not used or are not suitable for testing methods [33]. The fourth is the *AHP fuzzy comprehensive evaluation method*. This method, on the one hand, obtains subjective judgment data through expert evaluation, and on the other hand, uses mathematical methods for data processing, which can realize the unification of qualitative and quantitative evaluation [29]. The fifth is the *difference-in-difference method (DID)* that, in recent years, has been widely used in policy evaluation research, as it can effectively deal with selectivity bias [34]. The basic idea is to allow the influence of unobservable factors, which are time independent.

In recent years, scholars have increasingly adopted the *DID* method in research on policy impact assessments. For example, Yang et al. adopted the *DID* method to study whether the Air Pollution Prevention and Control Action Plan promulgated by China has had a positive effect on air pollution control [35]. Chen et al. regarded the "China Carbon Emissions Trading Pilot Policy" as a quasi-natural experiment and used the *DID* method to identify the net causal effect of this environmental policy on corporate innovation [36], finding that the implementation of this policy would significantly reduce corporate innovation in general, which is basically consistent with the causal effect of the EU Emissions Trading Scheme. Chabé-Ferret and Subervie evaluated the impact of French agricultural environmental policies on farmers' green agricultural production and consumption [37]. Rivers et al. estimated the policy impact of a single-use plastic bag tax in Toronto, Canada, and found that the tax increased the reuse rate of shopping bags by 3.4% [38]. Gehrsitz quantified the positive impact of low-emission zones on air quality and birth rates in Germany [39].

Existing research on the implementation effect of ecological compensation policy in great lake basins is in theory more qualitative, albeit with limited empirical analysis. The main methods used are the propensity score matching method, AHP fuzzy comprehensive

evaluation method, etc., but these methods are not ideal for to deal with endogeneity; the *DID* method can effectively address this problem. Therefore, in this study we focus on the Poyang Lake River Basin, the largest freshwater lake in China, and use the *DID* method to evaluate the implementation effect of the ecological compensation policy. We also provide a research basis for other performance evaluations of basin ecological compensation policies.

The rest of this paper is organized as follows: Section 3 provides a description of the study area; Section 4 outlines the research methods and data sources; Section 5 includes the empirical analysis of the implementation effect of the watershed ecological compensation policy; Section 6 outlines the robustness test; Section 7 is the conclusion and policy enlightenment.

## 3. Overview of the Study Area

Poyang Lake and Dongting Lake are the largest watersheds in Jiangxi Province and Hunan Province [40], respectively. Poyang Lake is the largest freshwater lake in China, located in the northern part of Jiangxi Province [41], on the south bank of the middle and lower reaches of the Yangtze River. The Poyang Lake River Basin includes five major water systems: Ganjiang, Fuhe, Xinjiang, Rao, and Xiuhe Rivers. The Dongting Lake is the second largest fresh water lake in China, located in the northern part of Hunan Province, also on the south bank of the middle and lower reaches of the Yangtze River. The Dongting Lake River Basin [42] includes four major water systems: Xiangjiang, Zishui, Yuanjiang, and Lishui River. The Poyang Lake River Basin covers the whole Jiangxi Province and flows through 11 prefecture-level cities [43], including Nanchang, Ji'an, Yichun, Ganzhou, Fuzhou, Yingtan, Shangrao, Jingdezhen and Yichun, and 100 counties under its jurisdiction. Among them, Huichang, Zhanggong, Quannan, Xinzhou, Zhushan Wuyuan, and Suichuan are the upstream counties of the Poyang Lake River Basin. The downstream counties of the Poyang Lake River Basin mainly include Xihu, Chaisang, Nanchang, Wannian, and Guixi, as shown in Figure 1. The Dongting Lake River Basin covers the whole Hunan Province, including 13 prefecture-level cities such as Changsha, Zhuzhou, Xiangtan, and Hengyang, and 124 counties under its jurisdiction. The upstream counties of the Dongting Lake River Basin [44] mainly include Fenghuang, Hongjiang, Shaoyang, Wugang, Ningyuan, Jiangyong, while the downstream counties of the Dongting Lake River Basin are mainly Furong, Yuelu, Changsha, Taojiang, Miluo, Huarong, and Yueyang, as shown in Figure 2.

In addition, it should be further explained that Jiangxi Province was included in the Ecological Civilization Pilot Demonstration Zone in 2014. The pilot project was carried out in accordance with the ecological compensation policy of the whole river basin in Jiangxi Province. As the pilot project in the central region of China, it provides reference and basis for the future implementation of the watershed ecological compensation policy for other provinces. Moreover, Jiangxi Province and Hunan Province are closely adjacent, and they are very similar in terms of geographical location and economic and social development. Therefore, Jiangxi Province has implemented an ecological compensation policy for the Poyang Lake Basin, while Hunan Province has no ecological compensation policy for the Dongting Lake Basin.

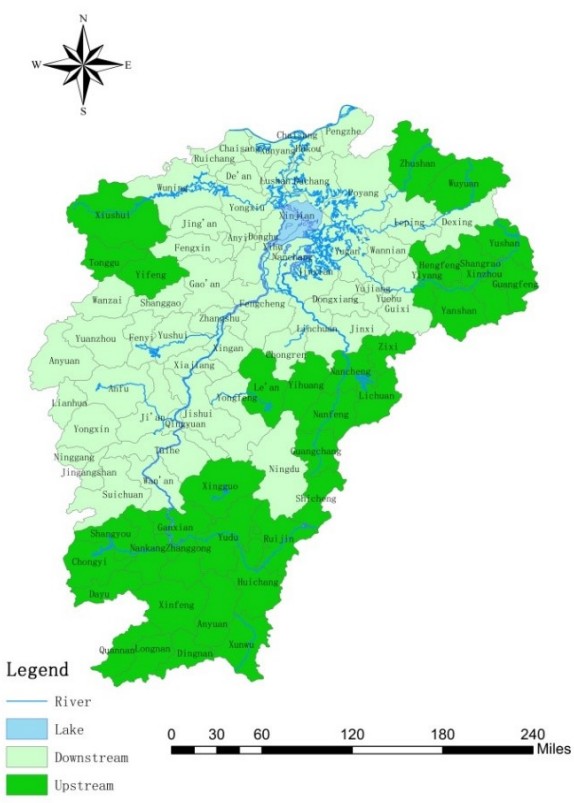

**Figure 1.** The upper and lower reaches of the Poyang Lake River Basin.

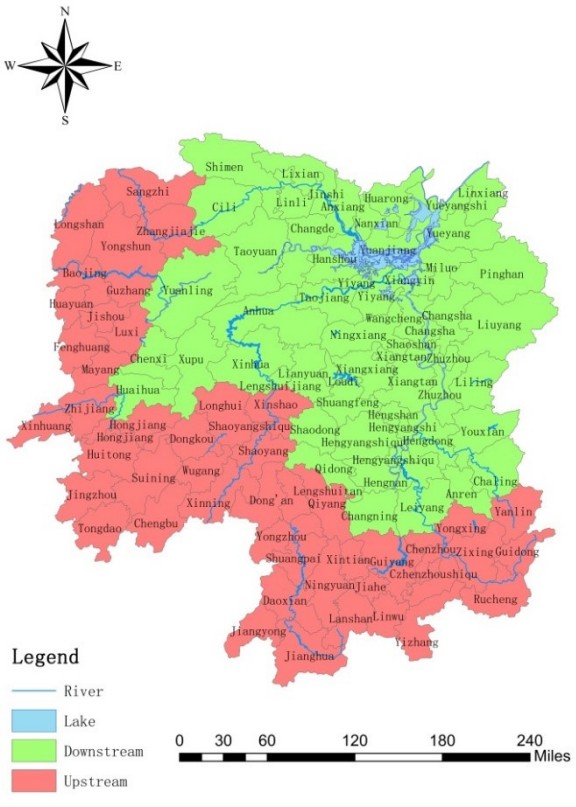

**Figure 2.** The upper and lower reaches of the Dongting Lake River Basin.

## 4. Research Methods and Data Sources

### 4.1. Research Methods and Variable Selection

DID is a method used for evaluation of policy impact, system performance evaluation, and project evaluation [45]. The concept of the DID method is that we usually choose regions or individuals that are not affected by the policy as the control group, and those regions or individuals that will be affected by the policy as the treatment group. Differences in the control group before and after the policy implementation can be seen as indicating the pure time effect, which when subtracted from the changes in the treatment group yields the "net effect" of the policy [46]. The quasi-natural experiment of the DID method can effectively avoid the problems of endogeneity and missing variables in the process of evaluating environmental policy effects [47].

#### 4.1.1. Variable Selection

Based on the *DID* method and reference literature combined with the actual situation of Poyang Lake, we designed one explained variable and three control variables. The specific variables are listed in Table 1.

**Table 1.** Variable construction.

| Variable Name | Variable Assignment | Variable Description | References |
|---|---|---|---|
| WQG (Water Quality Grade) | "Inferior class V" = 0, "class V" = 1, "class IV" = 2, "class III" = 3, "class II" = 4, "class I" = 5 | These explained variable represent the policy impact on water quality. | [28] |
| PCG (Per Capital GDP) | RMB100mn | The influence of water quality of the river basin on a per capita production level within the administrative area. | [29,32] |
| Area | Square kilometers | | [48] |
| Position | "Upstream" = 0, "Downstream" = 1 | The influence of watershed position on water quality. | [18,49] |

#### 4.1.2. Model Design

We analyze data from 224 counties in the Jiangxi and Hunan Provinces. The basic idea is to divide the data into two groups, one including the area (or individuals) not affected by the policy and are considered as the control group, and another including the policy implementation area and is considered the experimental group. The Poyang Lake River Basin was taken as the experimental group and the Dongting Lake River Basin as the control group.

The reference model is as follows:

$$WQG_{i,t} = \alpha_0 + \alpha_1 Policy_{i,t} \times Time_t + \gamma Y_{i,t} + \lambda_t + \mu_i + \zeta_{i,t} \tag{1}$$

In Equation (1), *WQG* (Water Quality Grand) is water quality grade, *i* represents regional virtual variable, *t* is water quality monitoring month, $Y_{i,t}$ is the control variable, $\lambda_t$ is the time fixed effect, $\mu_i$ is the regional fixed effect, $\zeta_{i,t}$ and is the random interference term. When the basin is in the Jiangxi Province, and the water quality monitoring month is January 2016 or later, the interaction term $Policy_{i,t} \times Time_t$ is equal to 1; otherwise, the interaction item is 0.

### 4.2. Data Source and Variable Description

The control variables are water quality, per capita GDP, and administrative area. The research period was from 2013 to 2018. The data on water quality mainly come from ***the national environmental quality standard for surface water*** (GB3838-2002), and ***Jiangxi Environmental Quality Monthly Report, Hunan Ecological Environment Status Bulletin***. The data on basin position, per capita GDP, and administrative area mainly

come from the statistical *yearbook of Jiangxi Province, statistical yearbook of Hunan Province, statistical yearbook of Nanchang City, statistical yearbook of Changsha City,* and *statistical yearbook of Jiujiang City*. Through data availability, validity, and data continuity, the discontinuous monitoring-section data before and after the implementation of the policy were deleted, and a total of 4248 observation data of 22 experimental groups and 37 control groups were obtained. The specific variables are listed in Table 2.

**Table 2.** Statistical table of variable description.

| Variable | Observations | Mean Value | Standard Deviation | Min. | Max | Median |
|---|---|---|---|---|---|---|
| WQG | 4248 | 3.680 | 0.638 | 0.000 | 5.000 | 4.000 |
| Policy × Time | 4248 | 0.186 | 0.390 | 0.000 | 1.000 | 0.000 |
| A-Area | 4248 | 1706.611 | 1243.151 | 84.000 | 4950.000 | 1592.500 |
| PCG | 4248 | 5.29 | 6.372 | 0.258 | 75.210 | 3.005 |
| Position | 4248 | 0.424 | 0.494 | 0.000 | 1.000 | 0.000 |

Based on the Stata16.0 platform, we analyzed the trend of average water quality before and after the implementation of the policy, and the results are shown in Figure 3.

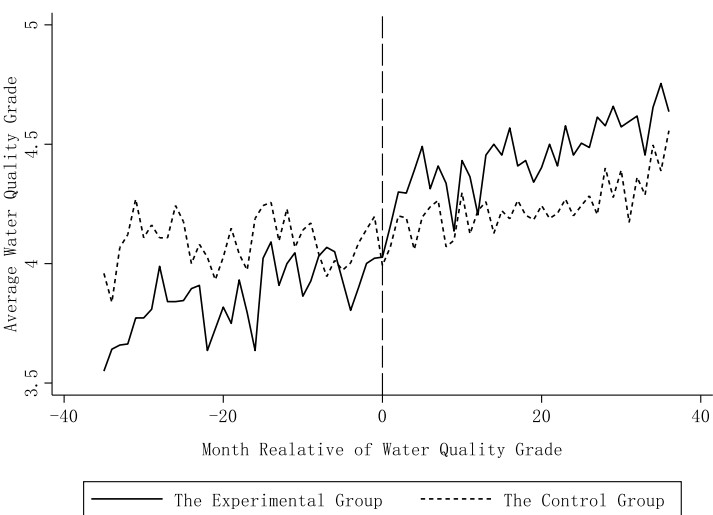

**Figure 3.** Average water quality change trend chart.

According to the results in Figure 3, the average water quality of the Poyang Lake River Basin was lower than that of the Dongting Lake River Basin before the implementation of the policy, while the average water quality of the Poyang Lake River Basin was higher than that of the Dongting Lake River Basin after the implementation of the policy. Meanwhile, we can also find that the average water quality of the Poyang Lake River Basin and the Dongting Lake River Basin have the same upward trend, so it is feasible to use the difference-in-difference mode in this study.

## 5. Policy Evaluation and Testing

### 5.1. Parallel Trend Test

Since the premise of the *DID* method is to satisfy the hypothesis of the parallel trend test, this study cancels the fixed time and uses the model to test the parallel trend. The reference model is as follows:

$$WQG_{i,t} = \alpha_0 + \alpha_1 Policy_{i,t-35} + \alpha_2 Policy_{i,t-34} + \alpha_3 Policy_{i,t-33} + \cdots\cdots + \alpha_{36} Policy_{i,t}$$
$$+ \alpha_{70} Policy_{i,t+34} + \alpha_{71} Policy_{i,t+35} + \alpha_{72} Policy_{i,t+36} + \gamma Y_{i,t} + \mu_i + \zeta_{i,t} \quad (2)$$

In Equation (2), Policy$_{i,t \pm n}$ is a dummy variable that represents the months before and after the implementation of the policy. The model takes the time *t* of policy implementation as the current

observation point to investigate the changes in water quality in 72 months before and after the implementation of the policy. In the regression results, if the coefficient of $Policy_{i,t\pm n}$ is not significant, it indicates that there is a parallel trend between the experimental group and the control group before the implementation of the policy. If the coefficient of $Policy_{i,t\pm n}$ is significant, the effect of ecological policy on water quality improvement is apparent. The trend of the estimated coefficient is shown in Figure 4. The horizontal axis represents the months before and after implementation of the policy, and the vertical axis represents the estimated value of the coefficient. Before the implementation of the policy, the coefficient of $Policy_{i,t\pm n}$ was not significant, indicating that there was no significant difference between the experimental and control groups. After the implementation of the policy, its coefficient was significantly positive, indicating that the policy has a positive impact on water-quality improvement, and its impact is persistent. At the same time, in the first half of the year before the implementation of the policy, the implementation of ecological protection improved the water quality to a certain extent. Therefore, in the year before the implementation of the policy, the estimated coefficient fluctuated around 0, which was significantly positive, and the effect coefficient after implementation was highly significant positive. According to the above analysis, the improvement of water quality is closely related to the promulgation of ecological compensation policy, and the results of the *DID* method are credible.

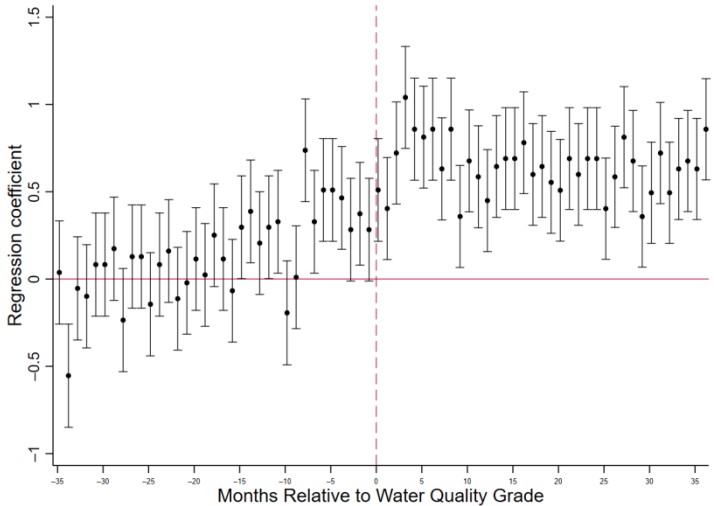

**Figure 4.** Parallel trend test chart.

Moreover, based on the stata16.0 platform, we conducted *t*-test on the water quality variables before the implementation of the policy, and the results are shown in Table 3.

**Table 3.** *t*-test results for the water quality grade before the implementation of the policy.

| Group | 0 | 1 |
|---|---|---|
| n | 1332 | 791 |
| Mean | 3.627 | 3.399 |
| Diff ! = 0 | 0.000 | 0.000 |
| P(T<t) | 1.000 | |

From the results in Table 3, we can find that the difference of water quality grade between the experimental group and the control group before the implementation of the policy is 0, indicating that the characteristics of the experimental group and the control group are similar, which is in line with the hypothesis of parallel trend test. Therefore, it further illustrates the feasibility of using the difference-in-difference method.

### 5.2. Model Operation Results

Based on the stata16.0 software platform, the monthly average water quality data is estimated by the *DID* method [50]. The basic estimation results are listed in Table 4. In Table 4, columns 1 and 3 are not fixed with area and time effects; column 1 and column 2 do not include control

variables. The estimation results of the model show that the coefficient $\alpha_1$ of the interaction term $Policy_{i,t} \times Time_t$ is highly significant regardless of whether the control variables are added. At the same time, it can be found that after adding control variables and controlling regional and temporal effects, the net effect coefficient of the basin ecological compensation policy on water quality and ecological environment improvement is 0.322, which is significantly positive at the 1% level. In addition, regional characteristics and economic conditions also have different impacts on local water quality. Since the geographical characteristics have obvious impacts on the ecological environment of water quality, the research results of this paper are analyzed based on Model (4) in Table 4.

**Table 4.** Difference-in-Difference Model estimation results.

| | Dependent Variable: Water Quality Grade | | | |
|---|---|---|---|---|
| | **(1)** | **(2)** | **(3)** | **(4)** |
| $Policy_{it} \times Time_t$ | 0.323 *** | 0.322 *** | 0.323 *** | 0.322 *** |
| | (0.000) | (0.000) | (0.000) | (0.000) |
| A-Area | | | 0.000 ** | −0.000 ** |
| | | | (0.023) | (0.009) |
| PCG | | | 0.002 | −0.005 |
| | | | (0.335) | (0.109) |
| Position | | Control | 0.164 *** | −0.248 * |
| | | | (0.000) | (0.062) |
| Province FE | | | | Control |
| Time FE | | Control | | Control |
| Constants | 3.327 *** | 3.471 *** | 3.556 *** | 3.780 *** |
| | (0.000) | (0.000) | (0.000) | (0.000) |
| Observations | 4248 | 4248 | 4248 | 4248 |
| $R^2$ | 0.092 | 0.311 | 0.103 | 0.312 |

Note: ***, **, and * represent significance at the 1%, 5%, and 10% confidence levels, respectively.

### 5.3. Analysis on Environmental Factors of Ecological Water Quality in the Poyang Lake River Basin

According to the results of "*DID* model estimation" (Table 4), there is a significant correlation between position, administrative area, and water-quality ecological environment. Among them, the location of the basin has a positive impact on the ecological environment of water quality, that is, the water quality in the upper reaches of the basin is better than that in the downstream. The counties located in the lower reaches of the river basin have larger "negative externalities" due to the geographical environment, and the water quality is poor. The area of the administrative region has a negative impact on the water quality and ecological environment of the basin because of high responsibility-subject implementation efficiency in a small geographical area. When the policy is implemented in a small area, the efficiency and effect of the government on the water quality in the basin are increased. In addition, the basic estimation results from Table 4 show that per capita GDP has a marginally significant negative correlation with the watershed quality, which indicates that the daily production activities of residents have a certain amount of impact on water quality pollution. At the same time, it can be found that there are significant differences between the regression results in the table after controlling for regional and time effects, which indicates that the implementation of the policy is closely related to the time and characteristics of the basin.

### 5.4. The Influence of per Capita GDP on Policy Implementation

The results in Table 4 show that the effect of per capita GDP on water quality has a marginally significant correlation. Therefore, we divided the per capita GDP into high and low parts according to the median, and studied the effect of ecological policy implementation when the regional population economic activities were different. The reference model is as follows:

$$WQG_{i,t} = \alpha_0 + \alpha_1 Policy_{i,t} \times Time_t + \gamma P_{i,t} + \lambda_t + \mu_i + \zeta_{i,t} \tag{3}$$

In Equation (3), $P_{i,t}$ refers to the variable after removing the non-highly significant variable from the control variable of Equation (1) as the control variable of this equation.

Table 5 shows the model regression results for different regions of per capita GDP. In the results, the first column is the regional estimation of the weak economic activity region, the second column is the regional estimation of strong economic activity, and the interaction coefficient is significantly

positive. At the same time, the effect of policy implementation in the region with weak economic activity is greater than that in the region with strong economic activity. This is mainly because areas with more economic activities cause greater water pollution, and the policy has a lower effect on the improvement of water quality.

**Table 5.** The effect of per capita GDP on policy.

| Variable | Dependent Variable: Water Quality Grade | |
| --- | --- | --- |
| | (1) | (2) |
| $Policy_{it} \times Time_t$ | 0.308 *** | 0.243 *** |
| | (0.000) | (0.000) |
| control variable | Control | Control |
| Province FE | Control | Control |
| Time FE | Control | Control |
| Constants | 2.966 *** | 3.844 *** |
| | (0.000) | (0.000) |
| Observations | 2137 | 2111 |
| R$^2$ | 0.281 | 0.373 |

Note: *** represent significance at the 1% confidence levels, respectively.

## 6. Robustness Check

### 6.1. PSM-DID Check

The premise of the *DID* estimation model is the randomness and common trend premise of the pilot. In Figure 4, the common trend hypothesis has passed, but the randomness of the pilot is difficult to guarantee. To solve the problem of sample selection, we adopted the *PSM-DID* method to conduct a robust test. In the actual operation process referring to the related research of year-by-year matching [49], we adopted the method of month-by-month matching to carry out a robustness test. First, the experimental group was used to match the control group. Second, the water quality data of the experimental group were used as dependent variables, and other control variables were used as matching variables for logit regression to obtain the propensity matching score. Finally, the *DID* test is carried out again according to the matching results. After matching, a total of 642 groups of 1284 data samples were matched. The regression results are shown in Table 6.

**Table 6.** *PSM-DID* Robustness test results.

| Variable | Dependent Variable: Water Quality Grade | |
| --- | --- | --- |
| | (1) | (2) |
| $Policy_{it} \times Time_e$ | 0.419 *** | 0.428 *** |
| | (0.000) | (0.000) |
| Province FE | | Control |
| Time FE | | Control |
| Constants | 3.704 *** | 3.543 *** |
| | (0.000) | (0.000) |
| Observations | 1284 | 1284 |
| R$^2$ | 0.151 | 0.443 |

Note: *** represent significance at the 1% confidence levels, respectively.

### 6.2. Exclusion Policy Exogenous Test

The assumption of the DID estimation model is that the operation of each county is not affected by water quality. However, the local government has strong autonomy while choosing the time and place of operation, and the actual effect of the policy may be overestimated. Therefore, it is necessary to exclude the impact of water quality on policy. On the basis of Equation (1), we add the average value of water quality before the implementation of the policy ($Pre\_WQG_{i,t=0}$) as a variable to investigate whether the policy is affected by the water quality before the implementation of the policy. The reference model is as follows:

$$WQG_{i,t} = \alpha_0 + \alpha_1 Policy_{i,t} \times Time_t + \beta Pre\_WQG_{i,t=0} + \gamma P_{i,t} + \lambda_t + \mu_i + \zeta_{i,t} \qquad (4)$$

Table 7 shows the estimated results after adding variable $Pre\_WQG_{i,t=0}$. The results show that the coefficient of the interaction term $\alpha_1$ is still significantly positive, consistent with the basic regression results. The regression results show that there is no obvious choice error in the basic model, which can eliminate the exogenous interference of policy. At the same time, it was found that after controlling the region and time, the influence of administrative area and basin position on water quality changes from positive to negative, which indicates that water quality has a strong correlation with region and time.

**Table 7.** Policy exogenous test results.

| Variable | Dependent Variable: Water Quality Grade | | | |
|---|---|---|---|---|
| | **(1)** | **(2)** | **(3)** | **(4)** |
| $Policy_{it} \times Time_t$ | 0.322 *** | 0.322 *** | 0.322 *** | 0.322 *** |
| | (0.000) | (0.000) | (0.000) | (0.000) |
| $Pre\_WQG_{i,t=0}$ | 0.635 *** | 1.900 ** | 0.627 *** | 1.158 ** |
| | (0.000) | (0.002) | (0.000) | (0.003) |
| A-Area | | | 0.0000 | −0.0002 ** |
| | | | (0.937) | (0.026) |
| PCG | | | 0.0401 * | −0.2158 * |
| | | | (0.065) | (0.086) |
| Position | | | −0.0010 | −0.0045 |
| | | | (0.641) | (0.109) |
| Province FE | | Control | | Control |
| Time FE | | Control | | Control |
| Constants | 1.326 *** | −3.337 | 1.347 *** | −0.558 |
| | (0.000) | (0.117) | (0.000) | (0.681) |
| Observations | 4248 | 4248 | 4248 | 4248 |
| $R^2$ | 0.254 | 0.311 | 0.255 | 0.312 |

Note: ***, **, and * represent significance at the 1%, 5%, and 10% confidence levels, respectively.

### 6.3. Placebo Test

The experimental group and the control group may be affected by other omissions, which can lead to errors in the evaluation of the policy effect. Therefore, we referred to the method of Shen and Jin [51], and made fictitious processing on the experimental and control groups. On the premise that the experimental group and the control group were known, the experimental and control groups were randomly selected for the placebo test [52]. In this experiment, the double difference estimation was repeated 1000 times based on random samples. The coefficient distribution is shown in Figure 5. From Figure 5, it can be seen that the coefficient is symmetrically distributed around 0, indicating that no other policy variables have been omitted [53].

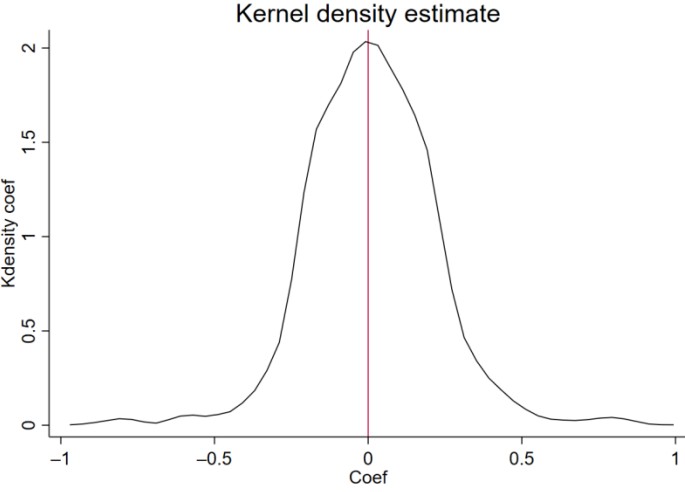

**Figure 5.** Placebo test results.

### 7. Policy Implications

Based on the empirical results, we proposed the following four policy implications, in order to better improve the watershed ecological compensation policy.

First, we adhere to the principle of ecological priority and carry out economic activities on the premise of ecological protection. The results showed that the implementation effect of regional policies with active economic activities was lower than that of regions with weak economic activities. Therefore, it is necessary to carry out regional economic activities at the expense of the environment. To counteract the negative effects, the government should strengthen the supervision of enterprises with negative externalities, build a reasonable ecological compensation mechanism, and promote the development of the regional economy under the premise of green ecology.

Second, we should pay more attention to the ecological compensation of the downstream areas according to local conditions. Due to the geographical location, the downstream area is in an ecologically inferior position. The behavior of the upstream area has a direct impact on the ecological water quality of the downstream area. Therefore, in implementing the ecological compensation policy, the principle of "compensation according to standard" and "compensation according to actual situation" should be carried out according to the ecological environment of downstream and upstream areas, and the standard of policy compensation should be improved.

Third, the local government should take the responsibility to improve the ecological supervision of large regional administrative regions. As far as water quality jurisdiction is concerned, the main regulatory bodies are mainly concentrated in local governments, and the impact of government regulation has significant positive effect on water quality. The empirical results show that the area of the administrative region has a significant impact on water quality, and the water quality under the jurisdiction of small community is generally higher. Therefore, in policy implementation, we should strengthen the supervision of the area of large regional administrative regions, the responsibility taken by the local government rather than of pursuance of the "performance" of economic development that imposes the cost on the ecological environment.

Fourth, we should invest more manpower and capital to strengthen the monitoring of the ecological environment. Some areas of the index statistics were not complete, the detection process was challenged by insufficient monitoring, the monitoring scope was incomplete, and so on. Therefore, under the current ecological goal, local governments should increase the input of technical elements and labor factors of ecological protection, and earnestly implement the strategic goal of ecological protection as the premise and high-quality economic development as the road.

### 8. Conclusions

This paper evaluated the watershed ecological compensation policy, as policy implementation standard through the double difference method. We examined whether the implementation of the policy to improve the water quality and ecological environment is effective in the case of Poyang Lake. The empirical results showed that the implementation of an ecological compensation policy has a significant effect on improving water quality. We found significant negative correlation between watershed location, administrative area, and water quality, and a marginally significant negative correlation between per capita GDP and water quality. We also found that the stronger the regional population economic activities, the weaker the effect of regional policy implementation.

**Author Contributions:** Model construction and measurement, Y.L. & L.H.; research design and methods, F.K.; constructive suggestions, C.X.; writing—review and editing, K.X.; questionnaire design and data collection, B.W. All authors have read and agreed to the published version of the manuscript.

**Funding:** This research was funded by the Special Project of Cultivating Leading Talents in Philosophy and Social Science of Zhejiang Province (21YJRC2ZD), 2019 Research Project of Soft Science Research Base for Water Security and Sustainable Development in Jiangxi Province (19JDYB07), 2020 Research Project of the Key Research Base for Humanities and Social Sciences in Jiangxi Province (JD20113).

**Informed Consent Statement:** Informed consent was obtained from all subjects involved in the study.

**Acknowledgments:** This research was supported by the Special Project of Cultivating Leading Talents in Philosophy and Social Science of Zhejiang Province (21YJRC2ZD), 2019 Research Project of Soft Science Research Base for Water Security and Sustainable Development in Jiangxi Province (19JDYB07), 2020 Research Project of Key Research Base for Humanities and Social Sciences in Jiangxi Province (JD20113).

**Conflicts of Interest:** The authors declare no conflict of interest.

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
