# Peer review of "Evaluation of the Implementation Effect of the Ecological Compensation Policy in the Poyang Lake River Basin Based on Difference-in-Difference Method"

_sustainability, doi:10.3390/su13158667_

Round 1

Reviewer 1 Report

The manuscript used DID method to analyze the Implementation effect of the 2

ecological compensation policy in the Poyang Lake,and got some results and gave some suggestions. It is interesting for ecological compensation research. But it is not clear in method and analyses. And it needs major revision.

Some comments:

  1. in the research, the Dongting Lake is designed as the control group. Please introduce the background of the two lakes. Why? The Dongting lake has no ecological compensation?
  2. For water quality improvement, the drive force is just ecological compensation? Not other factors? For example,environmental protection measures, Sewage collection and treatment?4. please analyze the water quality changes before and after carrying out ecological compensation policy.
  3. The results“the smaller the administrative area, the better the water quality in the basin” is not  scientific. Because big administrative area can be divided into more smaller administrative area.

Author Response

Comment 1: In the research, the Dongting Lake is designed as the control group. Please introduce the background of the two lakes. Why? The Dongting lake has no ecological compensation?

Responses: Thank you for your valuable comments. The details are as follows.

" Poyang Lake and Dongting Lake are the largest watersheds in Jiangxi Province and Hunan Province[40], respectively. Poyang Lake is the largest freshwater lake in China, located in the northern part of Jiangxi Province[41], on the south bank of the middle and lower reaches of the Yangtze River. The Poyang Lake River Basin includes five major water systems: Ganjiang, Fuhe, Xinjiang, Rao, and Xiuhe Rivers. The Dongting Lake is the second largest fresh water lake in China, located in the northern part of Hunan Province, also on the south bank of the middle and lower reaches of the Yangtze River. The Dongting Lake River Basin[42] includes four major water systems: Xiangjiang, Zishui, Yuanjiang, and Lishui River. The Poyang Lake River Basin covers the whole Jiangxi Province and flows through 11 prefecture-level cities[43], including Nanchang, Ji'an, Yichun, Ganzhou, Fuzhou, Yingtan, Shangrao, Jingdezhen and Yichun, and 100 counties under its jurisdiction. Among them, Huichang, Zhanggong, Quannan, Xinzhou, Zhushan Wuyuan, and Suichuan are the upstream counties of the Poyang Lake River Basin. The downstream counties of the Poyang Lake River Basin mainly include Xihu, Chaisang, Nanchang, Wannian, and Guixi, as shown in Figure 1. The Dongting Lake River Basin covers the whole Hunan Province, including 13 prefecture-level cities such as Changsha, Zhuzhou, Xiangtan, and Hengyang, and 124 counties under its jurisdiction. The upstream counties of the Dongting Lake River Basin[44] mainly include Fenghuang, Hongjiang, Shaoyang, Wugang, Ningyuan, Jiangyong, while the downstream counties of the Dongting Lake River Basin are mainly Furong, Yuelu, Changsha, Taojiang, Miluo, Huarong, and Yueyang, as shown in Figure 2.

In addition, it should be further explained that Jiangxi Province was included in the ecological civilization pilot demonstration zone in 2014. The pilot project was carried out in accordance with the ecological compensation policy of the whole river basin in Jiangxi Province. As the pilot project in the central region of China, it provides reference and basis for the future implementation of the ecological compensation policy of the river basin for other provinces. Moreover, Jiangxi Province and Hunan Province are closely adjacent, and they are very similar in terms of geographical location and economic and social development. Therefore, Jiangxi Province has implemented an ecological compensation policy for the Poyang Lake Basin, while Hunan Province has no ecological compensation policy for the Dongting Lake Basin."

Thank you very much.

Comment 2: For water quality improvement, the drive force is just ecological compensation? Not other factors? For example,environmental protection measures, sewage collection and treatment?

Responses: Thank you very much for your wonderful advice. The details are as follows.

 "Of course, these are other measures will greatly improve the water quality of the river basin, for example, environmental protection measures, sewage collection and treatment. But this study assumes that other factors remain unchanged, and only considers the impact of ecological compensation policies on water quality improvement In order to obtain a more ideal research effect, the Dongting Lake River Basin is used as the control group and the Poyang Lake River Basin as the experimental group, because they have a high degree of similarity. Therefore, this article adopts the DID to conduct empirical analysis on the effect of policy implementation, provided that other influencing factors remain unchanged. "

Thank you very much.

Comment 3: Please analyze the water quality changes before and after carrying out ecological compensation policy.

Responses: Thank you very much for your helpful advice. The details are as follows.

"Based on the Stata16.0 platform, we analyzed the trend of average water quality before and after the implementation of the policy, and the results are shown in Figure 3. "

Figure 3. Average water quality change trend chart.

"According to the results in Figure 3, the average water quality of the Poyang Lake River Basin was lower than that of the Dongting Lake River Basin before the implementation of the policy, while the average water quality of the Poyang Lake River Basin was higher than that of the Dongting Lake River Basin after the implementation of the policy. Meanwhile, we can also find that the average water quality of the Poyang Lake River Basin and the Dongting Lake River Basin have the same upward trend, so it is feasible to use the difference-in-difference mode in this study. "

Thank you very much.

Comment 4: The results“the smaller the administrative area, the better the water quality in the basin” is not  scientific. Because big administrative area can be divided into more smaller administrative area.

Responses: Thank you very much for your wonderful advice.

We have replaced "the smaller the administrative area, the better the water quality in the basin" with " areas with small administrative areas have a smaller population, which in turn leads to better water quality in the river basin ".

Reviewer 2 Report

The paper is good and I am sure it would be of great interest to ecologists, ecological modelers, and environmental planners alike. Ecological compensation is an important but difficult topic due to the lack of any standard. 
My comments are mostly about points that I believe should be clarified a bit better or something that maybe should be presented visually. 
(1) Validation: it is not clear to me how you performed validation, e.g. why not to show model predictions against data? This can be done in terms of probability distribution functions, avg trend etc...
(2) Global sensitivity and uncertainty analyses (GSUA) is missing and that wold be ideal to identify what factors matters the most considering their non-linear interactions (see e.g. the refs below). Is this GSUA dependent on local environmental features or not? Parameter and variable importance over space and time is important. 
(3) Visualization quality should be improved for sure by also providing some of the aforementioned topics. 

For the above motivations I suggest to accept the manuscript after Moderate/Major Revisions. The manuscript is good but I believe it can be even better / more clear after addressing the above comments. THX!

Pianosi et al. (2016)
Sensitivity analysis of environmental models: A systematic review with practical workflow
Environmental Modelling & Software
Volume 79, May 2016, Pages 214-232 

Convertino et al. (2014)
Untangling drivers of species distributions: Global sensitivity and uncertainty analyses of MaxEnt
Environmental Modelling & Software
Volume 51, January 2014, Pages 296-309

Author Response

Comment 1: Validation, it is not clear to me how you performed validation, e.g. why not to show model predictions against data? This can be done in terms of probability distribution functions, avg trend etc...

Responses: Thank you for your valuable comments. The details are as follows.

Part 1: " Based on the Stata16.0 platform, we analyzed the trend of average water quality before and after the implementation of the policy, and the results are shown in Figure 3. "

Figure 3. Average water quality change trend chart.

" According to the results in Figure 3, the average water quality of the Poyang Lake River Basin was lower than that of the Dongting Lake River Basin before the implementation of the policy, while the average water quality of the Poyang Lake River Basin was higher than that of the Dongting Lake River Basin after the implementation of the policy. Meanwhile, we can also find that the average water quality of the Poyang Lake River Basin and the Dongting Lake River Basin have the same upward trend, so it is feasible to use the difference-in-difference mode in this study. "

Part 2: "Moreover, based on the stata16.0 platform, we conducted t-test on the water quality variables before the implementation of the policy, and the results are shown in Table 3. "

Table 3. T-test results for the water quality grade before the implementation of the policy

Group

0

1

n

1332

791

Mean

3.627

3.399

Diff != 0

0.000

0.000

P(T<t)

1.000

"From the results in Table 3, we can find that the difference of water quality grade between the experimental group and the control group before the implementation of the policy is 0, indicating that the characteristics of the experimental group and the control group are similar, which is in line with the hypothesis of parallel trend test. Therefore, it further illustrates the feasibility of using the difference-in-difference method. "

Thank you very much.

Comment 2: Global sensitivity and uncertainty analyses (GSUA) is missing and that wold be ideal to identify what factors matters the most considering their non-linear interactions (see e.g. the refs below). Is this GSUA dependent on local environmental features or not? Parameter and variable importance over space and time is important. 

Responses: Thank you very much for your wonderful advice. GSUA is an excellent model, and it is ideal to identify what factors matters the most considering their non-linear interactions. However, this method may deviate a little from the focus of this research, and due to the timing of the revision of the paper, we will definitely make good use of this model in future research, in order to continuously improve the quality of the paper.

Thank you very much.

Comment 3: Visualization quality should be improved for sure by also providing some of the aforementioned topics.

Responses: Thank you very much for your helpful advice. We have improved the visualization quality of the figure 1 and figure 2, and we have provided some of the aforementioned topics. The details are as follows.

"Poyang Lake and Dongting Lake are the largest watersheds in Jiangxi Province and Hunan Province[40], respectively. Poyang Lake is the largest freshwater lake in China, located in the northern part of Jiangxi Province[41], on the south bank of the middle and lower reaches of the Yangtze River. The Poyang Lake River Basin includes five major water systems: Ganjiang, Fuhe, Xinjiang, Rao, and Xiuhe Rivers. The Dongting Lake is the second largest fresh water lake in China, located in the northern part of Hunan Province, also on the south bank of the middle and lower reaches of the Yangtze River. The Dongting Lake River Basin[42] includes four major water systems: Xiangjiang, Zishui, Yuanjiang, and Lishui River. The Poyang Lake River Basin covers the whole Jiangxi Province and flows through 11 prefecture-level cities[43], including Nanchang, Ji'an, Yichun, Ganzhou, Fuzhou, Yingtan, Shangrao, Jingdezhen and Yichun, and 100 counties under its jurisdiction. Among them, Huichang, Zhanggong, Quannan, Xinzhou, Zhushan Wuyuan, and Suichuan are the upstream counties of the Poyang Lake River Basin. The downstream counties of the Poyang Lake River Basin mainly include Xihu, Chaisang, Nanchang, Wannian, and Guixi, as shown in Figure 1. The Dongting Lake River Basin covers the whole Hunan Province, including 13 prefecture-level cities such as Changsha, Zhuzhou, Xiangtan, and Hengyang, and 124 counties under its jurisdiction. The upstream counties of the Dongting Lake River Basin[44] mainly include Fenghuang, Hongjiang, Shaoyang, Wugang, Ningyuan, Jiangyong, while the downstream counties of the Dongting Lake River Basin are mainly Furong, Yuelu, Changsha, Taojiang, Miluo, Huarong, and Yueyang, as shown in Figure 2. "

Figure 1. The upper and lower reaches of the Poyang Lake River Basin.

Figure 2. The upper and lower reaches of the Dongting Lake River Basin.

"In addition, it should be further explained that Jiangxi Province was included in the ecological civilization pilot demonstration zone in 2014. The pilot project was carried out in accordance with the ecological compensation policy of the whole river basin in Jiangxi Province. As the pilot project in the central region of China, it provides reference and basis for the future implementation of the ecological compensation policy of the river basin for other provinces. Moreover, Jiangxi Province and Hunan Province are closely adjacent, and they are very similar in terms of geographical location and economic and social development. Therefore, Jiangxi Province has implemented an ecological compensation policy for the Poyang Lake Basin, while Hunan Province has no ecological compensation policy for the Dongting Lake Basin. "

Thank you for providing us such great comments so that we can improve this work. The entire manuscript is revised in accordance with all reviewers’ comments and its readability is enhanced. Thank you very much.

Best regards,

Kai Xiong

Round 2

Reviewer 2 Report

The paper can be accepted. The authors have adequately revised the manuscript that is worth publication